# The Moderating Effect of the Sense of Power on Green (NonGreen) Appeal in Promoting Sustainable Consumption

Yue Ni [1] and Qiqi Cheng [2,*]

1    College of Media and International Culture, Zhejiang University, Hangzhou 310058, China
2    Duke-NUS Medical School, National University of Singapore, Singapore 169857, Singapore
*    Correspondence: qiqi.cheng@outlook.com

**Abstract:** Sustainable behavior could be promoted via green advertising. Based on the agentic–communal model and the construal level theory, this paper explores the moderating effect of sense of power on the effectiveness of green and nongreen appeals through a random experiment. We expect that in a powerlessness mindset, a green appeal outperforms a nongreen appeal and that in a power mindset, a nongreen appeal outperforms a green appeal with a reduced effect size. As expected, the results show that low-power consumers are more likely to be persuaded when the appeal emphasizes the green attribute rather than the nongreen attribute, whereas the converse holds (not significantly) for high-power consumers. The results also show a significantly positive effect of green appeal on WOM intention. The moderated serial multiple-mediator model indicates that attitudes toward the ad and brand serially mediate the effectiveness of advertising, which is moderated by power. Overall, those results demonstrate that the success of an appeal can be affected by psychological sense of power. The practical implications are also discussed.

**Keywords:** power; green advertising; green appeal; persuasion; construal level theory; agentic–communal model; mindset

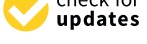

## 1. Introduction

Sustainable behavior that can ameliorate the environmental degradation caused by human activities could be promoted by advertising [1–3], such as recycling, energy saving, and water conservation [4,5]. Governments, companies, and political parties have extensively utilized green advertising to achieve their sustainable behavior-related objectives. How to improve the effectiveness of these advertisements is of significant importance to all parties involved. Previous studies indicate that a persuasive message is more powerful when tailored to an individual's unique psychological characteristics and motivations [6–9]. These findings suggest that psychological factors that influence how individuals process the information conveyed in the messages (for a review, see [10]) impact the persuasiveness of those messages. This pattern holds in studies of green advertising [7]. For example, the literature suggests that consumers' responses to green advertising are complex, and message recipients' preferences for one type of message over another are influenced by psychological factors such as abstract and concrete mindsets [11,12], and promotion focus and prevention focus mindsets [7]. In this line of research, the majority of the studies focus on exploring the moderating effect of psychological factors on the relative effectiveness of different appeals in green advertising (e.g., [2,13–18]). Only a few studies, however, investigate the moderating factors of the relative effects of green appeals versus nongreen appeals [7,19,20]. In this study, we aim to investigate the effect of the mindsets associated with high and low power (power mindset [21]) on the effectiveness of green versus nongreen appeals in advertising.

The power mindset is "a psychological orientation" [21,22] incited by structural differences in power, which is defined as "asymmetric control over valued resources in social

relationships" [23,24]. This psychological orientation, mindset, influences "the selection, encoding, and retrieval of information" and finally "drive evaluations, actions, and responses" [21,22]. Thus, it may moderate the persuasiveness of green advertising. In the literature, the terms "power," "power mindset," "sense of power," or "psychological state of power " are sometimes used interchangeably (e.g., [21,25]). Previous research suggests that a power (powerlessness) mindset would decrease (increase) perspective-taking [26], empathy [27], and generosity [28]; increase (decrease) psychological distance [29,30]; increase agentic (communal) orientation [31,32]; and increase (decrease) confidence [33]. Those consequences have a far-reaching effect on consumer behavior and message effects. Early research emphasized that the asymmetric position in social-economic status, social roles, or other aspects related to controlling valued resources would lead to power mindset [21]. Recent studies show that power mindset can also be incited independent of structural differences in power [33]. This development makes the power mindset even more important because a simple recall task can invoke this psychological orientation. During the past decade, increasing research has explored the effects of power on consumer behavior [31,34,35]. For example, relevant to this study, power has been found to be associated with green consumption [34,36,37].

There are also studies exploring how an audience's sense of power systematically influences the reception of persuasive messages. For example, one study found that a match between high (low) power and competence (warmth) information facilitates the persuasiveness of the message [6]. Another study found that a match between high (low) power and underdog (up-dog) appeals increased the message effect [38]. Power also moderates the effectiveness of gain-framed and loss-framed advertising messages [39]. However, the relationship between power and persuasion remains relatively unexplored. For example, to date, there is no study concerning whether a power mindset influences the effectiveness of green (vs. nongreen) appeals in advertising. Given the importance of power mindset and green adverting, the present research attempts to explore the moderating effect of power on the persuasiveness of green versus nongreen appeals to fill this gap.

One important mechanism by which the power mindset would influence the effectiveness of persuasive messages suggested in the literature is that mindsets bias the information used in evaluative judgment [6,21,40]. Previous studies find that a fit between consumers' mindsets and the content of persuasive messages enhances the message effect [10]. For example, a match between the promotion (prevention) focus mindset and nongreen (green) appeals [7], and a match between a high (low) power mindset and competence (warmth) information [6] increased the effectiveness of the advertising messages. In this study, drawing from the agentic–communal model and the construal level theory, we propose that a powerlessness mindset would value green appeal more and a power mindset would value nongreen appeal (competence in goal-related attributes) more. Moreover, a power mindset has other impacts on information processing that may influence the response pattern to persuasive messages. For example, compared with a powerlessness mindset, a power mindset will pay less attention to incoming information [40]. The abstract (concrete) mindset associated with the power (powerlessness) mindset will also focus more on the high- (low-) construal level information during the process of intaking information [41]. Those mechanisms may attenuate the difference between green and nongreen appeals in individuals with power mindsets. Thus, we expect that a powerlessness mindset would respond more positively to green appeals, whereas a power mindset would respond more positively to nongreen appeals, but with a reduced size effect.

The rest of this paper is organized as follows: First, we review prior research on power and powerlessness mindsets and develop testable hypotheses. We then test these hypotheses with a randomized online experiment. Finally, we discuss the theoretical and practical implications and suggest directions for future research.

## 2. Theoretical Background

### 2.1. Power and Its Effect in a Persuasive Context

The discrepancies between high-power and low-power individuals have been well documented in the literature. According to these studies, high-powered people are more self-focused and goal-focused [33,42,43] and thus evaluate those that are instruments of their goals more positively [44]. They are also more likely to ignore the social cost of action [35] and potential risks and threats related to it [35,45]. In contrast, low-power people are more inclined to consider the perspective of others. Thus, they are more likely to learn the social costs of their actions. To account for those discrepancies, scholars offered several theoretical explanations of power. Two theories of power, among them, may suggest possible mechanisms by which power influences the effectiveness of green advertising.

The first model is the agentic-communal model (ACM) of power [31,32]. Most of the studies regarding the influence of power on green consumption were based on this model [34,36,37]. The model postulates that high-power individuals adopt an agentic orientation and low-power individuals adopt a communal orientation [31]. With the agentic orientation, high-power people strive to individuate and expand the self and involve qualities such as efficiency, competence, and assertiveness that help attain their goals [46,47]. With communal orientation, low-power people strive to integrate themselves into some larger community of which they are part. Thus, they care for others and involve qualities such as benevolence, cooperativeness, and empathy, which are beneficial to being a part of the larger unit. The two orientations affect the type of information consumers attend to and value [6]. They focus on the information of the qualities valued by the two orientations, respectively, and shift the relative weight people place on the information about those dimensions. Therefore, the ACM of power suggests that high-power people would be more susceptible than their low-power counterparts to the arguments related to competence and efficiency in fulfilling their own goals (e.g., quality appeal). In contrast, low-power people would be more susceptible to the arguments related to communal benefits (e.g., green appeal).

The second model related is the construal level theory (CLT) of power, which postulates that a higher power induces a sense of psychological distance [29]. According to CLT, an object or event could be represented with high- or low-level construals (at a different level of abstraction) [48]. When the psychological distance is far, the abstract attributes that are the primary, essential attributes of the object (high-level construal) will be used to represent the object, whereas, when the psychological distance is close, the concrete attributes that are the secondary, peripheral characteristics (low-level construal) will be used to represent the object [48,49]. Therefore, the CLT of power suggests that power will influence how objects or events are mentally represented, with higher (lower) power leading to a more abstract (concrete) level of representation. The findings in CLT research also suggest that, in a higher construal level mindset, people will value those higher level construals more [49,50]. That is, when individuals are in a high-construal level mindset, they value the central attributes more, whereas when they are in a low-construal level mindset, the weight of peripheral attributes will be enhanced. Thus, an individual with a high-power mindset will value those high-level construals that are the central attributes of the products, and those with a low-power mindset will value those low-level construals that are the peripheral attributes. In the context of persuasion, previous studies suggest that a fit between the construal level of mindsets and the messages predicts an improved effect of the message. Thus, according to this theory, a match between high (low) power and high (low) construal level appeal (nongreen and green appeal) would enhance the message effect. The two theories suggest that the mindset induced by high or low power would determine which dimensions of the information would be valued and overweighted in a persuasive context.

### 2.2. Green Attribute and Appeal

Advertising could make claims about different attributes of a product. For example, green attributes, product-quality attributes, and other attributes linked to a favorable

product evaluation, could be utilized as appeals in advertising. For the green appeal, we refer to those appeals that link better environmental consequences with the consumption of the product in comparison to other alternatives [51,52]. In contrast, nongreen appeals could be any attribute unrelated to environmental consequences. In this study, a nongreen appeal refers to the competence in the attributes that have instrumental values for the designed purpose of the product. For example, in cleaning products, the appeal of a high-tech formula that could improve its cleaning ability and efficiency is a nongreen goal-relevant attribute, whereas the appeal of a natural plant formula that is eco-friendly is the green attribute. Green attributes differ from nongreen attributes in the following aspects and thus may be evaluated differently by consumers with power or powerlessness mindsets.

Firstly, from a CLT perspective [49], the green attribute of a product is the lower-level peripheral attribute compared with those goal-related functional attributes for most products that are not designed for improving the environment. The rationale is obvious. The essential attributes of a product should be those that enable the product to achieve its primary purpose [49]. All additional attributes that do not contribute to a product's intended function are considered secondary, or low-level, attributes. Previous studies suggested that a high-power individual adopted a high construal level mindset, and a low-power individual adopted a low construal level mindset [29,30]. Within the framework of CLT, we postulate that power mindsets moderate the effectiveness of green and nongreen appeals in advertising. The studies investigating the matching effect of persuasion attempts suggested that matching consumers' mindsets with the construal levels of the messages (message topics, designs) would lead to better persuasion (e.g., [7], for a review, see [10]). This is because a fit produces cognitive fluency, which will be used as a cue to enhance the effectiveness of the message within the heuristic system [53]. In addition, a higher construal level mindset will focus more on the central and essential information [41]. Thus, the effect of cognitive fluency on the effectiveness of the message will be attenuated in a higher level mindset. Therefore, we expect that in a powerlessness mindset, a green appeal would be more persuasive than a nongreen appeal, whereas, in a power mindset, a nongreen appeal would be better than a green appeal, but this effect would be diminished.

Secondly, the basic tenet of the ACM of power [31] is that high-power individuals focus more on self-interest than low-power individuals do, and low-power individuals view themselves as more dependent on others than high-power individuals do [28]. The two orientations influence how people judge the social cost of their actions, including the consequence on the environment. A sustainable environment could be regarded as a public good [54], and thus one's actions towards the environment would have consequences for others. The high-power people who focus on their goals could take advantage of consuming the product to fulfill their own goals and interests without considering the social costs. The low-power people in a communal orientation will take others into account [28] and thus will take this social cost on the environment into account and be more cautious about a product's green attributes. Therefore, we expect that green(nongreen) attributes will be more attended by a powerlessness (power) mindset.

Thirdly, according to the self-validation hypothesis, compared with those with a powerlessness mindset, individuals with a power mindset rely more upon their own thoughts and pay less attention to subsequent information [55]. Thus, a power mindset will make individuals less likely to be persuaded by the information in the messages. If both types of appeals were less likely to be attended to by individuals with a power mindset, the difference between green and nongreen appeals in effectiveness would be reduced. Furthermore, a power mindset will reduce message elaboration; thus, they may process all incoming information in the heuristic system [40] and thus believe the two appeals equally [56]. This will also result in a diminished difference between green and nongreen appeals in high-power individuals. Overall, the proposed framework in this study is illustrated in Figure 1.

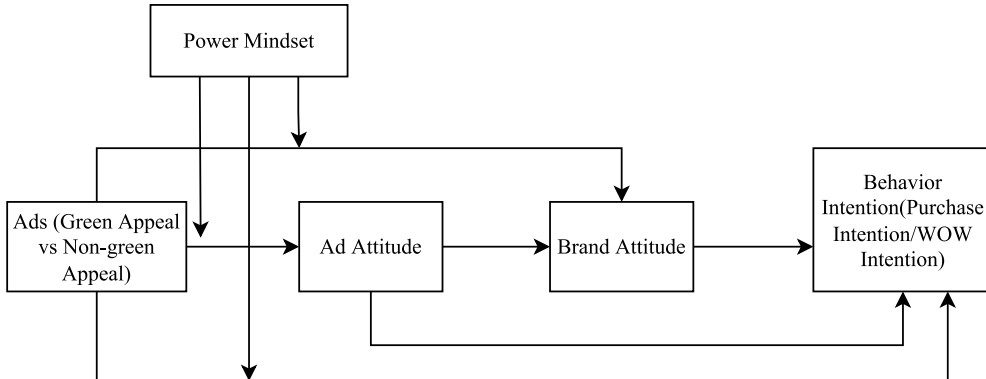

**Figure 1.** The Effect of Ads on Behavior intention.

In advertising effectiveness research, there are several operational measures of effectiveness, such as attitude toward the ad, attitude toward the brand, and purchase intention [57,58]. With marketers' interest in social marketing increasing, the effectiveness of advertising should also include the effect on the intention of giving word-of-mouth [59]; thus, we also measured the WOM intention. Therefore, we will test the following hypotheses on those four measurements of advertising effectiveness:

**Hypothesis 1:** *Significant interaction effects between appeal type (green and high-tech nongreen appeal) and mindsets (power and powerlessness) in determining (a) purchase intention, (b) WOM intention, (c) attitude toward the ads, and (d) brand attitude was expected.*

**Hypothesis 2:** *A green appeal versus a nongreen appeal will result in (a) increased purchase intention, (b) increased WOM intention, (c) a more favorable attitude toward the ad, and (d) a more favorable brand attitude in consumers with a powerlessness mindset.*

**Hypothesis 3:** *A nongreen appeal versus a green appeal will result in (a) increased purchase intention, (b) increased WOM intention, (c) a more favorable attitude toward the ad, and (d) a more favorable brand attitude in consumers with a power mindset. However, this effect is attenuated and may not be detected.*

Previous studies also suggest that attitude toward ads has a strong positive impact on attitude toward brand intention, as well as purchase intention [57,58], and that brand attitude has a positive impact on purchase intention [60,61]. Previous research has also indicated that attitudes toward advertisements and brand attitudes mediate the effect of advertising on behavior intention (e.g., [61]). Thus, we also have the following hypotheses:

**Hypothesis 4:** *There are positive relationships between (a) attitude towards ads and brand attitude; (b) attitude towards ads and purchase intention; (c) attitude towards ads and WOM intention; (d) brand attitude and purchase intention; (e) brand attitude and WOM intention; and (f) purchase intention and WOM intention.*

**Hypothesis 5:** *Attitude towards the ads and brand attitude mediate the effectiveness of the ads on (a) purchase intention and (b) WOW intention.*

## 3. Methods

### 3.1. Design and Participants

The main objective of this study is to examine whether an individual's power mindset moderates the effectiveness of advertising with green versus nongreen appeals. To test the proposed hypothesis, a 2 (Power mindsets: high vs. low) $\times$ 2 (Appeal type: green vs. nongreen) between-subject experimental design was employed. A total of 238 participants from the Credamo platform were recruited to participate in this study in exchange for a small

payment. Among those participants, 17 who did not understand or ignored the experimental manipulation (7%) were excluded. Thus, the final sample consisted of 221 participants ($M_{age}$ = 29.73, $SD_{age}$ = 9.07, Range = [19,62], 145 females).

### 3.2. Stimuli

To avoid the confounding influence of consumers' preexistent dispositions and attitudes toward a real brand due to familiarity [62], the ad stimuli used in the study featured a fictitious brand of household cleaning product (Clean Angel). Following Ku et al. [7], we drafted two ads in which the headlines communicated the green and nongreen appeals of the product. The nongreen appeal ad headline emphasized that the product was made from "high-tech formula" and that it was "efficient", whereas the green appeal ad headline emphasized that the product was made from "natural plant formula" and that it was "eco-friendly". The other elements in the two ads were the same, and both had "can remove tough stains and dirt with ease" following the headline (Appendix A). A cleaning product was selected since it is regularly used on a daily basis [11], and frequently appears in green commercials, which made the task more realistic [63].

### 3.3. Procedures

Power was manipulated with the procedure by Galinsky, Gruenfeld, and Magee [33]. Participants in the high-power treatments read a translated version of the following instruction in their native language: "Please recall a particular incident in which you had power over another individual or individuals. By power, we mean a situation in which you controlled the ability of another person or persons to get something they wanted or were in a position to evaluate those individuals. Please describe this situation in which you had power—what happened, how you felt, etc.". Those participants in the low-power treatments read the instructions in their native language: "Please recall a particular incident in which someone else had power over you. By power, we mean a situation in which someone had control over your ability to get something you wanted or was in a position to evaluate you. Please describe this situation in which you did not have power—what happened, how you felt, etc.".

After reading those instructions, participants were required to write down the incident. The contents they wrote were used to check whether they correctly understood the instructions for power manipulation. After the power manipulation, participants reported their subjective sense of power, which was used as a manipulation check [34].

Next, participants were informed that the study had entered stage 2, where they were first asked to read an advertisement and report their responses on the scales. Participants first read the green or nongreen appeal ads at this stage. They reported their purchase intention and word-of-mouth(WOM) intention afterward. Next, they reported their attitudes towards the advertising and the brand. After completing these scales, participants were asked to answer manipulation check questions. Finally, participants answered demographic questions and were debriefed.

### 3.4. Measures

Purchase intention. Participants' purchase intentions were measured by three items ("How likely/willing/inclined are you to purchase this clean product after viewing this ad") on a 9-point scale (1 = "extremely unlikely/unwilling/not inclined," and 9 = "extremely likely/willing/inclined"). The mean scores on the scales served as the measure of purchase intention ($\alpha$ = 0.90; M = 6.84, SD = 1.24).

WOM intention. WOM intention was measured by participants' responses to three statements ("I am likely to spread positive word of mouth about this clean product; I would recommend this clean product to my friends; If my friends were looking to purchase clean products, I would recommend this one.") on a 9-point scale (1 = Strongly disagree, 9 = Strongly agree). The scales were adopted from Maxham and Netemeyer [64], and

the mean of the scores on the scales served as the measure of WOM intention (M = 6.43, SD = 1.39; $\alpha$ = 0.88).

Ad attitude and brand attitude. Ad attitude and brand attitude were assessed using a 9-point, 4-item semantic differential scale, adopted from Zhang [65] and Bellman et al. [66]. For brand attitude, the bipolar ends were bad/good, unfavorable/favorable, unappealing/appealing and unlikeable/likeable ($\alpha$ = 0.87; M = 6.92, SD = 1.21). For ad attitude, the bipolar ends were unpleasant/pleasant, unlikeable/likeable, irritating/not irritating and not interesting/interesting ($\alpha$ = 0.92; M = 6.84, SD = 1.34).

## 4. Results

### 4.1. Manipulation Check

For the power manipulation check, the participants reported their subjective sense of power on three items (1 = not powerful at all/not at all in control/not influential at all, 7 = very powerful/in complete control/completely influential) after the power manipulation [34]. Participants' responses on the three items were consistent and thus were averaged to serve as an index of power ($\alpha$ = 0.95; M = 4.28, SD = 1.81). An ANOVA on this index indicates that there is a significant main effect of the high- or low-power treatment (M low = 2.78, M high = 5.77, $F_{(1, 219)}$ = 468.06, $p < 0.001$, η2 = 0.681). Therefore, the power manipulation was successful.

For the green and nongreen ad manipulation check, participants were asked to answer the following two questions on a 9-point scale (1 = none and nine = quite a lot): (1) "To what extent does the ad emphases the formula of the clean product is eco-friendly?" and (2) "To what extent, the ad emphases the formula of the product is high-tech?" Participants who read the green ad reported a higher score on the eco-friendly scale (M green group = 7.12, M nongreen group = 4.00, $F_{(1, 219)}$ = 160.83, $p < 0.001$, $η^2$ = 0.423), whereas participants who read the nongreen ad reported a higher score on the high-tech scale (M green group = 4.88, M nongreen group = 7.05, $F_{(1, 219)}$ = 85.24, $p < 0.001$, $η^2$ = 0.280). Therefore, the message manipulation was successful.

### 4.2. Purchase and WOM Intention

We conducted an ANOVA with purchase intention as the dependent variable, power status(1 = high power, 0 = low power), ad appeal (1 = nongreen appeal, 0 = green appeal), and the interaction between power and ad appeal as independent variables. The results showed that neither the main effects of the power ($F_{(1, 217)}$ = 2.16, $p = 0.144$, $η^2$ = 0.010) nor the ad appeal ($F_{(1, 217)}$ = 1.50, $p = 0.149$, $η^2$ = 0.010) were significant. As expected, the interaction was significant ($F_{(1, 217)}$ = 16.35, $p < 0.001$, $η^2$ = 0.070), indicating that H1a is supported (Figure 2A). For high-power participants, they reported a marginally significantly higher purchase intention in a nongreen high-tech appeal condition than in a green appeal condition (M green ad = 6.75, M nongreen ad = 7.16, $F_{(1, 217)}$ = 3.38, $p = 0.067$, $η^2$ = 0.015), suggesting that H3a is supported; for low-power participants, green versus nongreen appeal had a significantly larger effect on purchase intention (M green ad = 7.16, M nongreen ad = 6.28, $F_{(1, 217)}$ = 15.01, $p < 0.001$, $η^2$ = 0.065), suggesting that H2a is supported. Comparing the simple effects of power within the two ad conditions, for green appeal, the effectiveness in the low-power participants was marginally significantly higher than that in the high-power groups ($F_{(1, 217)}$ = 3.30, $p = 0.071$; $η^2$ = 0.015). However, for nongreen appeal, the effectiveness in the high-power groups was significantly higher than that in the low-power groups ($F_{(1, 217)}$ = 15.25, $p < 0.001$; $η^2$ = 0.066).

An ANOVA with WOM intention as the dependent variable yielded a significant main effect of appeal type ($F_{(1, 217)}$ = 3.89, $p = 0.050$, $η^2$ = 0.018), with green appeal gaining more WOM intention. The main effect of power was not significant ($F_{(1, 217)}$ = 0.304, $p = 0.582$, $η^2$ = 0.001). As expected, the interaction effect was significant ($F_{(1, 217)}$ = 10.29, $p = 0.002$, $η^2$ = 0.045), indicating that H1b is supported. For high-power participants, nongreen high-tech appeal was slightly better than the green appeal, but not significant (M green ad = 6.37, M nongreen ad = 6.60, $F_{(1, 217)}$ = 0.77, $p = 0.382$, $η^2$ = 0.004), supporting H3b.

For low-power participants, the green appeal ad gained a significantly higher effect than nongreen appeal on WOM intention (M $_{green\ ad}$ = 6.86, M $_{nongreen\ ad}$ = 5.91, F(1, 217) = 13.36, $p$ < 0.001, $\eta^2$ = 0.058), suggesting that H2b is supported. Comparing the simple effects of power within the two ad conditions, for the green appeal, the WOM intention in the low-power individuals was marginally significantly higher than that in the high-power groups (F(1, 217) = 3.51, $p$ = 0.062; $\eta^2$ = 0.016). However, for the nongreen high-tech appeal, the WOM intention in the high-power individuals was higher than that in the low-power individuals (F(1, 217) = 13.06, $p$ = 0.008; $\eta^2$ = 0.032), significantly. See Figure 2B.

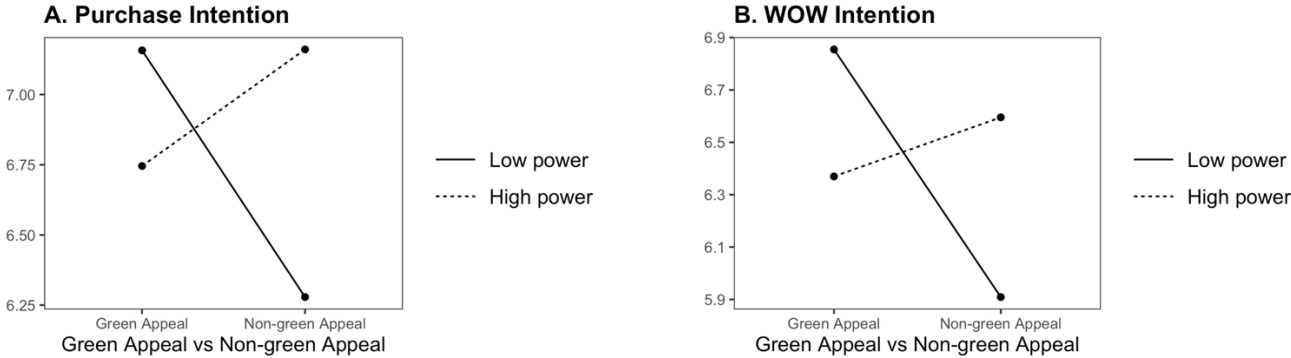

**Figure 2.** Interaction Effects on Purchase and WOW intention: Power × Green Appeal.

### 4.3. Ad Attitude and Brand Attitude

For the attitude toward the ads, an ANOVA showed that neither the main effects of the power (F(1, 217) = 0.079, $p$ = 0.779, $\eta^2$ = 0.000) nor the main effect of the ad appeal (F(1, 217) = 2.21, $p$ = 0.139, $\eta^2$ = 0.010) were significant. As expected, the interaction was significant (F(1, 217) = 9.64, $p$ = 0.002, $\eta^2$ = 0.043), indicating that H1c is supported (Figure 3A). For high-power participants, nongreen appeal was slightly better than green appeal, but the difference was not significant (M $_{green\ ad}$ = 6.81, M $_{nongreen\ ad}$ = 7.07, F(1, 217) = 1.31, $p$ = 0.253, $\eta^2$ = 0.006), supporting H3c. For low-power participants, green ads had a significantly better effect than nongreen appeal on ad attitude (M $_{green\ ad}$ = 7.26, M $_{nongreen\ add}$ = 6.53, F(1, 217) = 10.49, $p$ = 0.001, $\eta^2$ = 0.046), supporting H2c. Comparing the simple effects of power within the two ad conditions, for the green appeal, the ad attitude formed in the low-power individuals was significantly higher than that in the high-power groups (F(1, 217) = 3.97, $p$ = 0.048; $\eta^2$ = 0.018). However, for nongreen appeal, the ad attitude formed in the high-power groups was significantly higher than that in the low-power groups (F(1, 217) = 5.76, $p$ = 0.017; $\eta^2$ = 0.026).

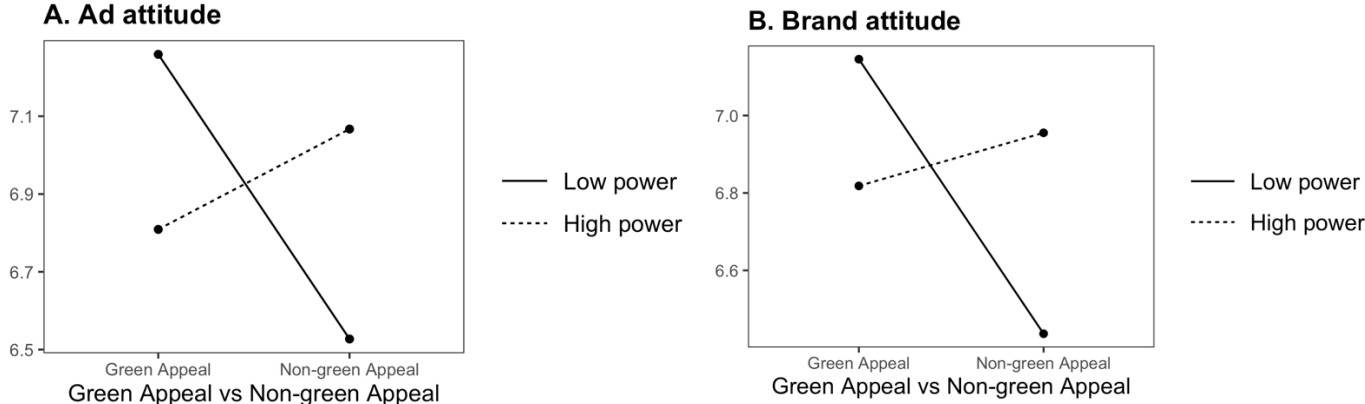

**Figure 3.** Interaction Effects on Ad and Brand Attitude: Power × Green Appeal.

For the brand attitude, the ANOVA resulted in no significant main effects for the power (F(1, 217) = 0.288, $p$ = 0.592, $\eta^2$ = 0.001), no significant main effect for the ad appeals

($F_{(1, 217)}$ = 2.56, $p$ = 0.111, $\eta^2$ = 0.012), but a significant interaction effect ($F_{(1, 217)}$ = 5.61, $p$ = 0.019, $\eta^2$ = 0.025), as expected in H1d (Figure 3B). For high-power participants, nongreen appeal was better than green appeal, but the difference was not significant (M $_{green\ ad}$ = 6.82, M $_{nongreen\ ad}$ = 6.96, $F_{(1, 217)}$ = 0.30, $p$ = 0.587, $\eta^2$ = 0.001), supporting H3d. For low-power participants, green ads had a significantly better effect on brand attitude than nongreen appeal (M $_{green\ ad}$ = 7.15, M $_{nongreen\ ad}$ = 6.44, $F_{(1, 217)}$ = 7.84, $p$ = 0.006, $\eta^2$ = 0.035), supporting H2d. For the green ads, there was no difference in the brand attitude formed in the high- and low-power participants ($F_{(1, 217)}$ = 1.67, $p$ = 0.198; $\eta^2$ = 0.008). However, for nongreen appeals, the brand attitude formed in the high-power groups was significantly better than that in the low-power groups ($F_{(1, 217)}$ = 4.24, $p$ = 0.041; $\eta^2$ = 0.019).

*4.4. Mediation Analysis*

For hypothesis H4, we identified significant correlations between ad attitude and brand attitude (*cor* = 0.89, $p < 0.001$), between ad attitude and purchase intention (*cor* = 0.84, $p < 0.001$), between brand attitude and purchase intention (*cor* = 0.81, $p < 0.001$), between ad attitude and WOW intention (*cor* = 0.86, $p < 0.001$), between brand attitude and WOW intention (*cor* = 0.85, $p < 0.001$), and between WOW intention and purchase intention (*cor* = 0.84, $p < 0.001$). H4a,b,c,d,e,f was supported. Thus, a moderated serial multiple mediator model was used to test this possible mechanism in our data. We utilized the PROCESS macro v4.1 for R (model 85, [67]) to estimate the moderated serial mediation of purchase intention (Figure 4) with 5000 bootstrap samples. We entered the purchase intention as the dependent variable, ad appeal (1 = nongreen appeal, 0 = green appeal) as the independent variable (X), ad attitude (M1) and brand attitude (M2) as serial mediators, and the power (1 = high power, 0 = low power) as the moderator (W), respectively. This model allowed us to test (a) the specific indirect effect through ad attitude, (b) the specific indirect effect through brand attitude, and (c) the indirect effect through ad attitude and brand attitude in serial, thus taking into account the positive relationship between the two variables.

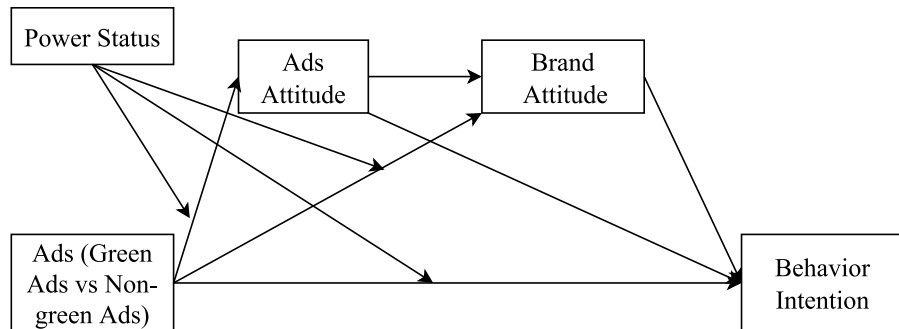

**Figure 4.** Moderated serial multiple mediator model (Process 4.1, Model 85) depicting the indirect effect of ads on behavior intention (purchase intention and WOM intention) through ad attitude and brand attitude, moderated by power status.

As expected, the index of moderated serial mediation via ad attitude and brand attitude was significant ($\beta$ = 0.254, CI95% = [0.063, 0.548]), supporting H5a. Within low-power individuals, the serial effect was significant ($\beta$ = −0.188, CI95% = [−0.390, −0.052]). This effect not significant within high-power individuals at the 95% level ($\beta$ = 0.066, CI95% = [−0.047, 0.222]). The index of moderated mediation via ad attitude was also significant ($\beta$ = 0.578, CI95% = [0.186, 1.079]). Within the low-power individuals, ad attitude partially mediated the effect of ads on purchase intentions ($\beta$ = −0.427, CI95% = [−0.768, −0.149]). This was not present in the high-power individuals ($\beta$ = 0.151, CI95% = [−0.107, 0.449]). The index of moderated mediation via brand attitude was not significant at 95% level ($\beta$ = −0.036, CI95% = [−0.131, 0.063]). For both the high ($\beta$ = −0.031, CI95% = [−0.092, 0.037]) and low ($\beta$ = 0.005, CI95% = [−0.058, 0.074]) power participants, brand attitude did not par-

tially mediate the effect. Including the significant indirect effect, a nongreen versus a green appeal also had a significant negative direct effect on purchase intention in participants with a low power mindset ($\beta = -0.27$, CI95% = $[-0.513, -0.025]$), suggesting that, compared with a green appeal, a nongreen appeal has a relatively smaller effect. In participants with mindset, the effect was positive, but not significant ($\beta = 0.23$, CI95% = $[-0.010, 0.468]$). See Figure 5 for all path coefficients at the two platforms.

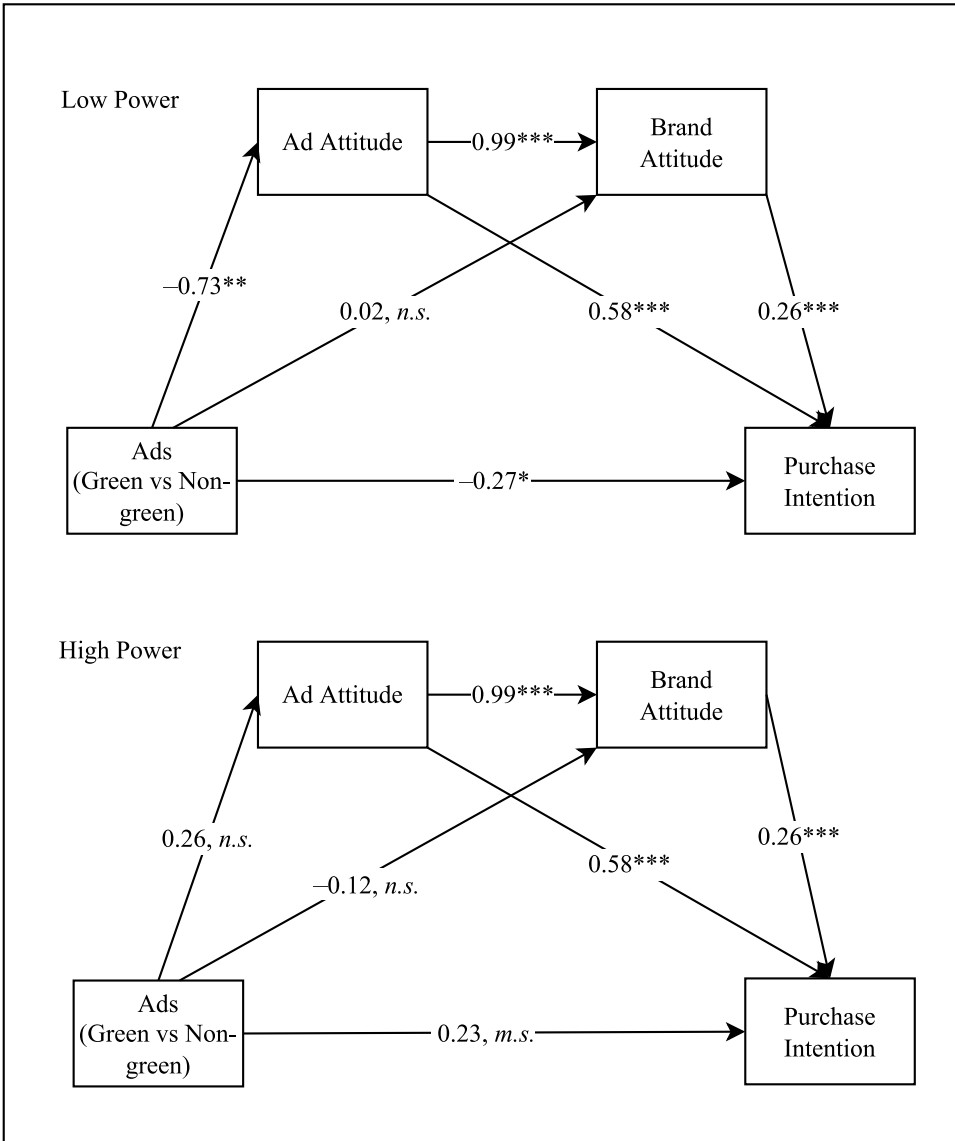

**Figure 5.** Moderated serial mediator model for purchase intention. Statistical significance is indicated by *n.s. p* $\geq$ 0.1, *m.s. p* < 0.1, * *p* < 0.05, ** *p* < 0.01, *** *p* < 0.001.

The same mediation analysis was applied to the WOM intention, and the results revealed a similar pattern. As expected, the index of moderated serial mediation via ad attitude and brand attitude was significant ($\beta = 0.416$, CI95% = $[0.155, 0.727]$), supporting H5b. Within low-power individuals, the serial effect was significant ($\beta = -0.307$, CI95% = $[-0.534, -0.124]$). This effect was not significant within high-power individuals at the 95% level ($\beta = 0.108$, CI95% = $[-0.081, 0.302]$). The index of moderated mediation via ad attitude was also significant ($\beta = 0.550$, CI95% = $[0.197, 0.969]$). Within the low-power individuals, ad attitude partially mediated the effect of ads on purchase intentions ($\beta = -0.407$, CI95% = $[-0.689, -0.154]$). This was not the case in the presence in the high-power individuals ($\beta = 0.143$, CI95% = $[-0.108, 0.414]$). The index of moderated mediation via brand

attitude was not significant at 95% level (β = −0.058, CI95% = [−0.204, 0.093]). For both the high (β = −0.050, CI95% = [−0.152, 0.054]) and low (β = 0.008, CI95% = [−0.090, 0.107]) power participants, brand attitude did not partially mediate the effect. In both low power (β = −0.239, CI95% = [−0.495, 0.016]) and high power (β = 0.024, CI95% = [−0.225, 0.274]) participants, the direct effects were not significant. See Figure 6 for all path coefficients at the two platforms.

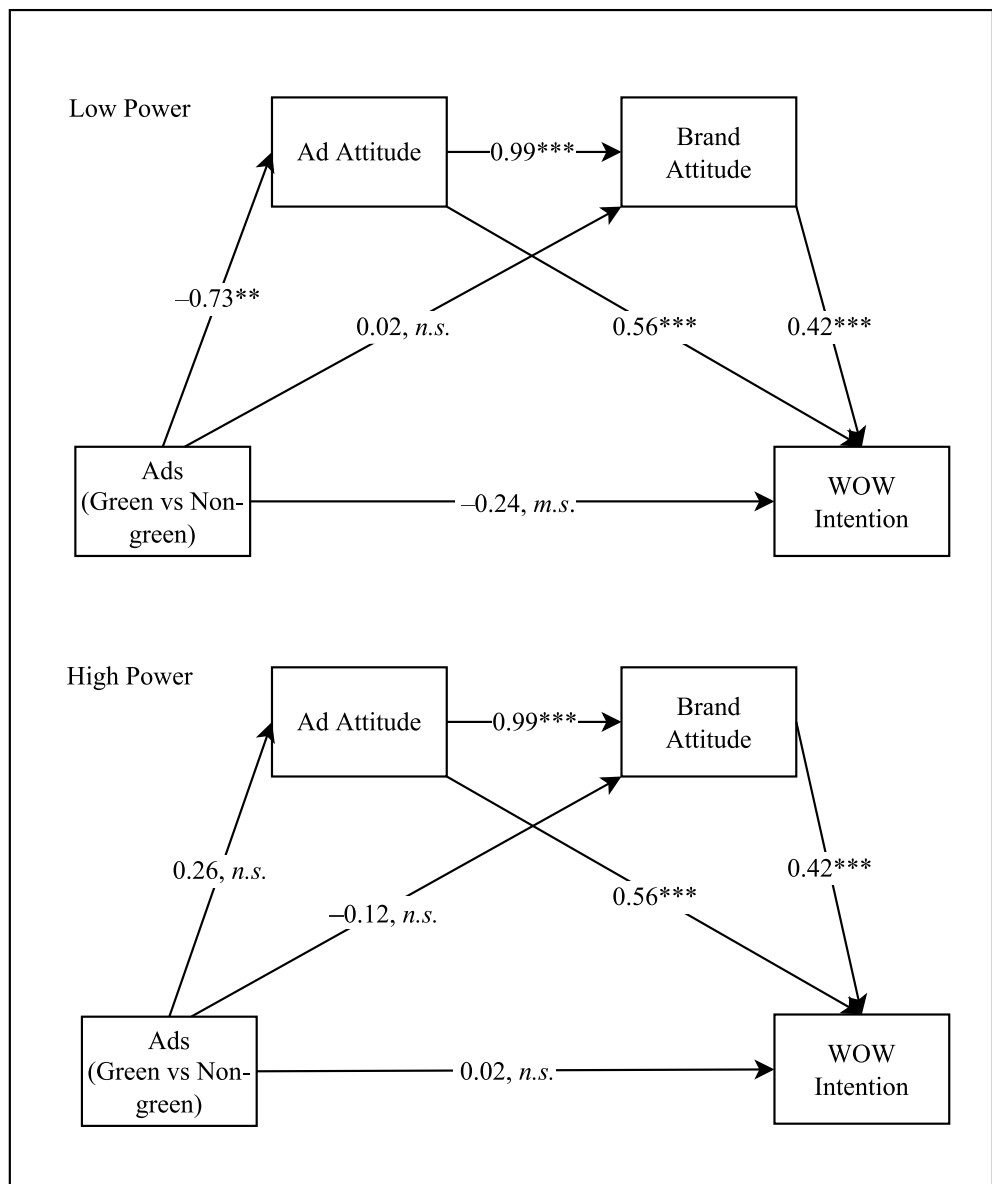

**Figure 6.** Moderated serial mediator models for WOW intention. Statistical significance is indicated by *n.s.* $p \geq 0.1$, *m.s.* $p < 0.1$, ** $p < 0.01$, *** $p < 0.001$.

## 5. Discussion

### 5.1. Summary of the Findings

This study presents an initial attempt to test the moderating effects of a sense of power on the effectiveness of green versus nongreen appeals in advertising. Our study adopted the perspective of the agentic-communal model of power [31], which organizes the basis of the motives determining consumers' goals and actions in terms of a focus on either agentic or communal orientation, and the CLT, which postulates that a high power is associated with a high construal level mindset, as the frameworks for analyzing consumers' responses to green over nongreen appeals. Based on those frameworks, this current study proposes that

consumers with a powerlessness (power) mindset would appreciate the green (nongreen) appeal in an ad; hence, they favor a green (nongreen) appeal ad over a nongreen (green) appeal ad. However, this effect will be reduced for consumers with a power mindset. The reasons are as follows: First, the abstract mindset associated with a power mindset allows them to focus on the central and positive arguments of different appeals in ads [49,50], which would reduce the effect of cognitive fluency in persuasion [41,68]. Second, the power mindset will induce confidence and rely more on their thoughts and beliefs, or the power mindset will lead to less elaboration in message scrutiny [40], and thus treat different appeals equally [56].

This study tested the predicted hypotheses by employing a randomized online experiment. The results showed that, as expected, in low-power consumers, the effectiveness of a message linking product attributes to a green appeal was greater than a message linking them to a nongreen appeal (supporting H2). The participants with a powerlessness mindset reported higher purchase intention (H2a), increased WOM intention (H2b), more favorable ad attitude (H2c), and more favorable brand attitude (H2d) in the green appeal condition than they did in the nongreen condition. In contrast, in high-power consumers, the converse emerged, although not significantly (H3). Those with a power mindset reported higher purchase intention (H3a), WOM intention (H3b), ad attitude (H3c), and brand attitude (H3d) in the nongreen appeal condition than in the green appeal condition. All those effects were positive, although not significantly. Therefore, the findings support the moderating effect of power and powerlessness on consumer evaluations of green versus nongreen advertising appeals (H1). These results are consistent with the existing literature on the construal level theory [49,50].

The results also showed that, as expected, there were positive relationships between any two of the four ways to measure how effective advertising is. Moderated serial multiple mediator models of purchase intention and WOM intention revealed that the effectiveness of ads on the two types of behavior intentions was mediated via ad attitude and brand attitude, which is moderated by the power status. Those findings are consistent with previous studies investigating the relationship between ad attitude, brand attitude, and behavior intention [57,58,61,69]. We add to the literature by adding WOM intention in this relationship. In addition, the result also indicates that the green appeal had a significant main effect in promoting WOM intention, suggesting that people would be more likely to share green appeal ads rather than nongreen appeal ads on average.

The literature in persuasion research suggests that the audience's power mindset may influence the process of persuasion through several paths (for a review, see [40]). Firstly, a power mindset will bias the thinking directions in the elaboration. For example, Dubois, Rucker, and Galinsky [6] suggested that power and powerlessness mindsets are biased in how they weigh competence and warmth arguments. In the current study, our results support the proposed notion that a power mindset puts more weight on a nongreen appeal, whereas a powerlessness mindset places more weight on a green appeal. Secondly, a power mindset will result in less elaboration in message scrutiny [40]. This effect will reduce the effectiveness difference between green and nongreen appeals in high-power consumers, for they simply believe the information conveyed in the advertising [56]. In this case, the effect originating from the biased thinking directions will be reduced in high-power consumers, which could explain the insignificant effect in the power mindset conditions. Thirdly, if the elaboration level is high, a power mindset will make people rely more on current thoughts and less likely to be persuaded by new information [40]. This path will also reduce the difference between green and nongreen appeals in high-power consumers because, in both conditions, they are less likely to be persuaded by the two appeals.

From a CLT perspective, consumers in a high-level power mindset process information about the high-level nongreen attributes more fluently, and those who are in a low-level powerlessness mindset process low-level green appeal more fluently. The feeling of cognitive fluency originating from a match between the mindsets and the information content in terms of construal level could influence the judgment as a non-essential input. This type of

incidental non-essential input has a larger influence on consumers with a low-level mindset because this detailed information weighs more in a low-level mindset [41]. Therefore, consumers may bias the information inputs according to their high- or low-level mindsets, and high-level mindsets will reduce this judgment bias by focusing on the essential high-level inputs. This study shows that participants with a power mindset reported a non-significant difference when evaluating nongreen and green appeals.

## 5.2. Theoretical Implications

Although many studies have applied the agentic-communal power model in exploring green consumption behavior in various contexts, little is known about the relative persuasiveness of green versus nongreen appeals for individuals whose power differs. This paper adds to the extant literature in the following ways: First, this study introduces a new moderator for studying the effectiveness of green versus nongreen appeal in advertising by demonstrating that the effectiveness of green versus nongreen appeal varies as a function of psychological sense of power. Previous research in green advertising has revealed several factors that influence the effectiveness of green advertising [70–73]. However, no studies have explored the moderating effect of the sense of power. This paper has filled the gap. Second, this study advances the notion that marketing message evaluations are a function of the compatibility or matching of consumers' power mindsets with the attributes in the appeals. Previous studies have suggested that a fit (matching) between the mindset of an individual and the messages presented would induce better persuasion (for a review, see [10]). This study adds to this line of research by exploring the effect of the power mindset [21]. The results support the general prediction of the matching effect of mindsets. Third, this study could also be classified as an extension of CLT research. This study advances the CLT studies by emphasizing the relative construal levels of green versus nongreen attributes of a product. Based on this assumption, future research could explore the moderating effect of other high- or low-construal level factors on the effectiveness of green versus nongreen appeals.

## 5.3. Managerial Implications

This study offers evidence that participants whose power had been experimentally manipulated do respond differently to green and nongreen appeals in advertising. The findings have significant practical implications for marketing managers and policymakers to build green strategies through the use of power, which contributes to sustainable development.

First, this study provides new ideas for market segmentation in green marketing. Companies and policymakers can deliver green or nongreen appeals to individuals with different power mindsets to promote sustainable behavior more effectively. For example, through membership systems, companies can collect information on the chronic power mindset of their customers, which can be used to determine whether to advertise green or nongreen appeals. The ad can also trigger the person's temporary power mindset (e.g., [28]). This means that the ad can trigger the low-power or high-power mindsets before delivering green or nongreen appeals to make the appeals more effective.

Second, this study offers new insight into the use of advertising to encourage sustainable behavior. The findings imply that nongreen appeals can also be used to promote green behavior and eco-friendly consumption. The majority of prior green marketing strategies have focused on a product's green features, which is likely to reduce consumers' perception of the product's nongreen attributes. According to this study, we can target advertising to those with a high or low sense of power based on other competitive features of the product or the sustainable behavior it promotes. For those with a high sense of power, green appeals may not be needed to promote sustainable behavior; for those with a low sense of power, green appeals may be better delivered with more specific, low-interpretation level messages.

Third, given the finding that green (vs. nongreen) appeal was more effective in activating WOM intention, if a marketing manager wants to try eWOM marketing, a green

appeal should be deployed. Currently, eWOM marketing is becoming more important than ever. Therefore, if this kind of marketing strategy were to be considered, the positive effect of a green appeal on WOM intention should be noted.

### 5.4. Limitations and Prospective Future Research

This study has investigated the impact of consumers' high and low sense of power on the effects of green and nongreen appeals in advertising. It has several limitations, and future work should address those limitations.

First, in this experiment, we only included a cleaning product. Previous studies suggest that product type moderates the effectiveness of green appeals [73]. Although the cleaning product is one of the most widely used products on a daily basis and in green advertising research [11,15,63,74–78], the findings based on this product may not apply to other products. For example, the green signaling hypothesis suggests that a signaling benefit of green consumption would lead to a green premium [79]. Some green products or sustainable behavior may prime high-power consumers with new motivations, such as signaling by consuming. When signaling is the reason for purchasing, the instrumental value of the product is no longer the previously designed one but signaling to high-power individuals. Thus, in this case, the green versus nongreen appeals may have a positive effect on high-power individuals. Future studies can test the moderating effect of a power mindset on the effectiveness of green versus nongreen appeals with different types of products.

Second, in this study, we employed two simple gain-framed ad messages. In practice, ad messages could be created with different appeals, topics, and designs, which would influence the construal level of the messages (for a review, see [10]). Within the CLT framework, it is expected that a power mindset would moderate the effectiveness of the ads with different construal levels. Thus, a power mindset is expected to moderate gain-framed or loss-framed messages, distant or proximal temporal framed messages, distant or proximal spatial framed messages, and narrative or non-narrative messages. Those designs could also be used in green advertising. Thus, future studies should empirically test whether the power mindset plays a moderating role in evaluating green advertisements created with those designs.

Third, although power has been found to be negatively associated with perspective-taking robustly, the relationship between the two is not invariant. For example, power can increase perspective-taking in cases where power activates a sense of responsibility for others [80,81]. Therefore, those moderators are expected to influence the effect of a power mindset on green appeal. Future work could test whether a sense of responsibility for others moderates the effect of a power mindset on green and nongreen appeals.

## 6. Conclusions

To conclude, this study explored the moderating effect of a psychological sense of power on the effectiveness of green versus nongreen appeals in advertising. The results showed that low-power consumers (powerlessness mindset) are more likely to be persuaded when the appeal emphasizes green attributes rather than nongreen high-tech attributes, whereas the converse holds for those high-power people with a reduced effect size, which is not identified in the current study. The results suggest that the effectiveness of the green versus nongreen appeal varies as a function of the psychological sense of power. The moderated mediation analysis further shows that the effectiveness of green versus nongreen appeal is mediated by the attitude towards the advertising and brand, which is moderated by the psychological sense of power. In addition, the results also showed a significant main effect of green appeal on WOM intention, indicating that consumers are more likely to share the product with others if a green appeal is utilized. Overall, these results demonstrate that the success of a green versus nongreen appeal can be affected by the psychological sense of power.

**Author Contributions:** Conceptualization, Y.N. and Q.C.; Data curation, Y.N.; Formal analysis, Y.N. and Q.C.; Funding acquisition, Q.C.; Methodology, Y.N.; Project administration, Y.N.; Supervision, Q.C.; Validation, Q.C.; Visualization, Y.N. and Q.C.; Writing—original draft, Y.N.; Writing—review & editing, Q.C. All authors have read and agreed to the published version of the manuscript.

**Funding:** This research was supported by the Fundamental Research Funds for the Central Universities (Social Science Experimental Research Youth Fund Project, Zhejiang University), grant number 2018QNA125.

**Institutional Review Board Statement:** Not applicable.

**Informed Consent Statement:** Informed consent was obtained from all subjects involved in the study.

**Data Availability Statement:** The datasets analyzed in this study are available from the corresponding author upon reasonable request.

**Conflicts of Interest:** The funders had no role in the design of the study, in the collection, analysis, or interpretation of data, in the writing of the manuscript, or in the decision to publish the results.

## Appendix A  Green and Nongreen Ads

| Green Appeal | Nongreen Appeal |
| --- | --- |
| 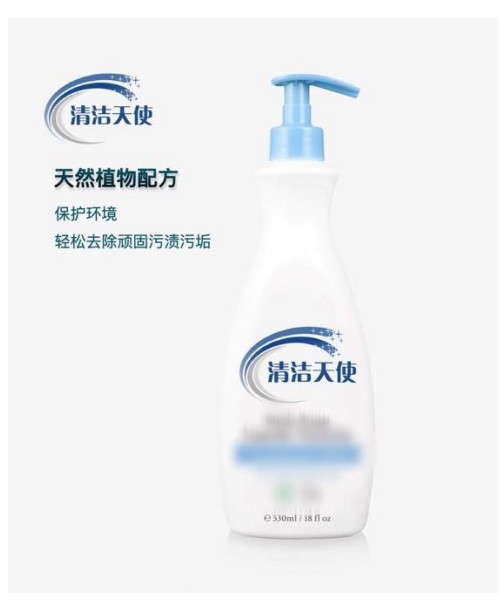 | 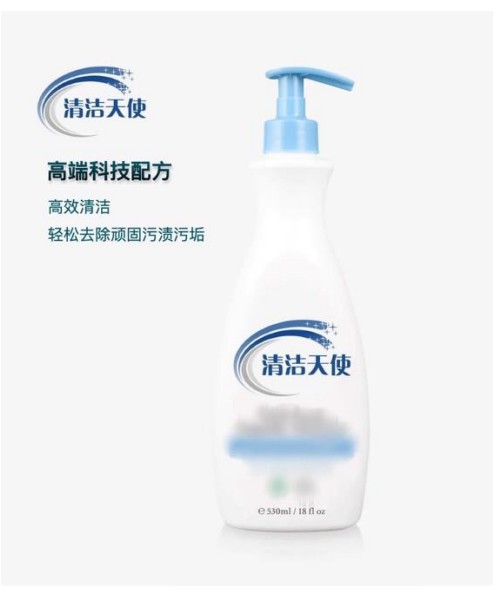 |
| Clean Angel (The brand)<br>• Natural plant formula<br>• Eco-friendly<br>• Can remove tough stains and dirt with ease | Clean Angel (The brand)<br>• High-tech formula<br>• Efficient cleaning<br>• Can remove tough stains and dirt with ease |

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
