# Peer review of "The Moderating Effect of the Sense of Power on Green (NonGreen) Appeal in Promoting Sustainable Consumption"

_sustainability, doi:10.3390/su142012983_

Round 1
Reviewer 1 Report
Basically this research is interesting and have significant findings. The further suggestions are as follows:
1) It would be better to introduce "power mindset" earlier instead of "power" in the introduction section;
2) The introduction part should be more relevant with the research background, research necessity and research question. The first part now reads more like an extended abstract and should be improved.
3)Is power mindset related to self-centered?
4) The recommendation is too general. For now, it seems that the empirical results makes sense, but there also need some innovative thinkings and practical recommendations.
Reviewer 2 Report
Your work has merit, however, there are some issues you need to address:
1. Line 81, which references are [23-2]
2. introduction, change the order, there is a misunderstanding in lines 71 y72, when you write "next", it seems from the previous one, which is not.
3. introduction, you need to write an introduction to the issue, not the results: erase from 60 to 68, instead, emphasize why the academy needs your research and where is the gap that you might fulfil and why you choose cleaning products.
4. In the results, what do you mean by B? do you mean beta? please find the appropriate symbol
5. Please, the results in figure 5, we can identify some negative results, which normally means the opposite results, How do you describe this result? In fact, ads in low power mindset have a negative impact on Ad attitude and purchase intention.
6. Maybe you need to create more hypotheses, because a hypothesis is accepted or not, not it depends on. It seems a research question rather than a hypothesis. Please, write specific hypotheses. Otherwise, it is very confusing for the reader. Once you change, change the whole article accordingly.
Round 2
Reviewer 1 Report
The authors have clearly addressed all the comments. The minor change needed is to provide translation for advertising words in appendix A.
Author Response
Dear reviewer,
Thank you for bringing this point to our attention. We have added the English translation of the original ads written in Chinese to the appendix, as suggested.
Regards,
The authors
Reviewer 2 Report
The article has really improved and you have addressed all the issues.
Author Response
Dear reviewer,
In this version, we added the English translation of the original ads written in Chinese to the appendix, as suggested by Reviewer 1. Thank you again for your very valuable comments and suggestions.
Regards,
The authors